

# A Spatiotemporal Weighted Regression Model (STWR v1.0) for Analyzing Local Non-stationarity in Space and Time

Xiang Que [1,2], Xiaogang Ma[2,*], Chao Ma[2,*], Qiyu Chen[3]

[1] Computer and Information College, Fujian Agriculture and Forestry University, Fuzhou, Fujian, China

[2] Department of Computer Science, University of Idaho, 875 Perimeter Drive MS 1010, Moscow, ID 83844-1010, USA

[3] School of Computer Science, China University of Geosciences (Wuhan), 388 Lumo Road, Wuhan 430074, China

*Correspondence to*: Xiaogang Ma (max@uidaho.edu); Chao Ma (chao@uidaho.edu)

**Abstract.** Local spatiotemporal non-stationarity occurs in various natural and socioeconomic processes. Many studies have attempted to introduce time as a new dimension into the geographically weighted regression model (GWR), but the actual results are sometimes not satisfied or even worse than the original GWR model. The core issue here is a mechanism for weighting effects of both temporal variation and spatial variation. In many geographical and temporal weighted regression models (GTWR), the concept of time distance has been inappropriately treated as time interval. Consequently, the combined effect of temporal and spatial variation is often inaccurate in the resulting spatiotemporal kernel function. This limitation restricts the configuration and performance of spatiotemporal weights in many existing GTWR models. To address this issue, we propose a new spatiotemporal weighted regression (STWR) model and the calibration method for it. A highlight of STWR is a new temporal kernel function, in which the method for temporal weighting is based on the degree of impact from each observed point to a regression point. The degree of impact, in turn, is based on the rate of value variation of the nearby observed point during the time interval. The updated spatiotemporal kernel function is based on a weighted combination of the temporal kernel with a commonly used spatial kernel (Gaussian or bi-square) by specifying a linear function of spatial bandwidth versus time. Three simulated datasets of spatiotemporal processes were used to test the performance of GWR, GTWR and STWR. Results show that STWR significantly improves the quality of fit and accuracy. Similar results were obtained by using real-world data for the precipitation hydrogen isotopes ($\delta^2$H) in Northeastern United States. The Leave-one-out cross-validation (LOOCV) test demonstrates that, comparing with GWR, the total prediction error of STWR is reduced by using recent observed points. Prediction surfaces of models in this case study show that STWR is more localized





than GWR. Our research validates the ability of STWR to take full advantage of all the value variation of past observed
points. We hope STWR can bring fresh ideas and new capabilities for analyzing and interpreting local spatiotemporal non-
stationarity in many disciplines.
**Key words:** Geographical and temporal weighted regression; Geographically weighted regression; Temporal non-
stationarity; Spatial analysis; Spatiotemporal variations; Spatiotemporal weighted regression.
**1 Introduction**
Time, space and attributes are three essential characteristics in geographic entities, and they are recorded to reflect the state
and evolution of various real-world phenomena and processes. Because space and time frame all aspects of the discipline of
geography (Goodchild, 2013), it is important to observe the spatiotemporal variations and explore appropriate analytical
methods to study and reason the internal mechanisms and evolutionary laws. In recent years, new platforms and instruments
have brought increasingly massive spatiotemporal data, such as the time- and geo-tagged sensor monitoring records and
remote sensing images. Those big data create great opportunities for studying human and environmental dynamics from
different perspectives, such as the patterns of human behavior (Chen et al., 2011), environmental risk assessment (Sun et al.,
2015), and disease outbreaks (Takahashi et al., 2008). Nevertheless, although spatiotemporal modeling has been a long-term
research focus in the field of geographical information science (GIScience) (Cressie, 1991; Cressie and Wikle, 2015), the
models are not mature yet and challenges still exist (Fotheringham et al., 2015), which call for further work.

In this paper, the technological development and discussion focus on modeling local spatiotemporal variations within

the framework of geographically weighted regression (GWR). GWR is a method for modeling spatially heterogeneous
processes (Brunsdon et al., 1996, 1998; Fotheringham et al., 2003). It has been applied in a variety of areas, such as climate
science (Brown et al., 2012), geology (Atkinson et al., 2003), mineral exploration (Wang et al., 2015), transportation analysis
(Cardozo et al., 2012), crime studies (Cahill and Mulligan, 2007; Wheeler and Waller, 2009), environmental science (Mennis
and Jordan, 2005), and house price modeling (Fotheringham et al., 2015). GWR calibrates a separate regression model at
each location through a data-borrowing scheme, in which distance-weights can be calculated by drawing on data from



neighboring observations of each regression point (Fraser et al., 2012). This operation complies with Tobler's first law of
geography - "Everything is related to everything else, but near things are more related than distant things" (Tobler, 1970).
Numerous studies have been devoted to incorporating the temporal dimension into spatial regression (Pace et al., 2000;
Gelfand et al., 2004; Crespo et al., 2007; Cressie and Wikle, 2015). However, most of these studies assume that temporal
effects are constant over space from a global perspective of modeling (Fotheringham et al., 2015). To address that issue,
Crespo et al. (2007) extended GWR by developing spatiotemporal bandwidths that account for varying local spatial effects
across time. Huang and Wu (2010, 2014) proposed a geographical and temporal weighted regression model (GTWR) with a
method of measuring the spatiotemporal 'closeness' and a parameter ratio $\tau$ to deal with different measured units in time
and space. Although the approach can address the issue to some extent, Fotheringham et al. (2015) pointed out that a sole
measurement of integrated spatial and temporal distances can be misleading as location and time are usually measured at
different scales, and he stated that the calculation of distance in three dimensions (time and two-dimensional space) remains
a challenge.
A spatiotemporal kernel function, which consists of mixed spatial and time-decay bandwidths, was proposed by
Fotheringham et al. (2015). Nevertheless, the stepwise strategy applied in this function for bandwidth optimization does not
always seem reasonable. In practice, this function needs to first find and fix an optimized spatial bandwidth, then it will find
the optimized temporal bandwidth. After that, the spatiotemporal weight will be calculated. This stepwise search process
means that the function is not able to optimize both temporal and spatial bandwidths at the same time. However, a more
reasonable thought is that the spatiotemporal bandwidth and its weight are simultaneously affected by both spatial and
temporal effects of a process. There should be ways to further improve the spatiotemporal kernel function in Fotheringham
et al. (2015).
The aim of this paper to develop a better methodology for the spatiotemporal kernel function. Following Tobler's first
law, we propose an algorithm, the spatiotemporal weighted regression (STWR). In STWR, the velocity of value change is
higher related if they were in near time and space. Therefore, STWR can borrow data not only from nearby locations, but
also from nearby value variation through time. The latter is what we call as "time distance" in STWR. The time distance is
not the concept of time interval, but the rate of value variation through time. It is a kind of value change that reflects the


temporal effect of nearby points to the regression point. Accordingly, our local spatiotemporal regression analysis model can
take advantage of the variation in data to identify temporal non-stationarity, which is an advantage comparing with GWR
and GTWR.

Before giving more details about STWR, we can further clarify the meaning of a few concepts. A common issue in the

existing GTWR models is that they use the concept of time interval, instead of the above-mentioned "time distance", to
calculate temporal and spatiotemporal weights. A time interval is the period between two observed time stages. A time
distance, in the context of STWR, is the rate of value variation between an observed point and a regression point through a
time interval. We can think about the following scenario for a group of points. The values of some points do not change or
change slightly from time A to time B, while a few other points may change greatly in that period. However, many GTWR
models ignore the difference in the value changes of observed points during a period of time, and regard that all these points
have the same temporal effect to their neighbor regression point. It is hard to believe that some unchanged observations
constantly affect their nearby regression points during the observed time interval. Intuitively, different variations of the
observed points have different temporal effects. For example, the faster the house price of a point change, the stronger the
temporal effect is to the house price at its nearby point. Moreover, the rate of value changes at different observed points
(time non-stationary) may also have spatial heterogeneity. The data values observed at different points are results of mixed
spatiotemporal effects and some other unknown factors (including errors). Therefore, using only time interval in the
calculation of temporal and spatiotemporal weights might interpret local spatiotemporal effect imprecisely.

There are other issues in the temporal kernel functions and the multiplication form of spatial and temporal kernels used

by the existing GTWR models (Huang et al., 2010; Wu et al., 2014; Fotheringham et al., 2015). When calculating the
spatiotemporal effect, these models generally use time intervals and the common kernel functions to calculate temporal
weights, such as Gaussian kernel or bi-square kernel. However, an appropriate temporal kernel function should not be the
same as the spatial kernel function, because space is in two or three dimensions while time is in one dimension and one
direction. Each regression point can borrow observed points from any directions in space but only use points from the past
rather than from the future. Moreover, these models directly calculate the integrated spatiotemporal weights by using a
multiplication of the spatial and temporal weights. For example, if the temporal effect weight 0.1, and the spatial effect





weight is 0.9, then those models will generate a spatiotemporal weight of 0.09. Such a simple multiplication may cause
underestimation of weights. For instance, to calculate the impact (weight) of the historical house price of B on the current
house price of A, there can be many possibilities. One is that if the overall house price changes quickly, then the historical
price of B may have little effect on the price of A, and the weight will be small. Another possibility is that the house prices
of locations around A have not changed much during a long period, then the historical price of B may still have a relatively
big impact on the current price of A. In this case, the weight will be seriously underestimated if the multiplication form of
space and time weights is used.
The above-mentioned limitations and issues in GWR and GTWR are the driving force behind our development of
STWR. The remainder of this article is organized as follows. Section 2 introduces the STWR model formulation, including
temporal kernel and spatiotemporal kernel functions. Section 3 describes the methods for bandwidth selection and calibration
when STWR is in operation. Section 4 presents results of applying GWR, GTWR and STWR to three sets of simulated data.
Section 5 presents experiment results with real-world precipitation hydrogen isotope data. In Section 6, we close the article
with a summary of the key findings and a few thoughts for future research.

## 2 The Core Model of STWR

### 2.1 The strategy of time distance decay

Since GWR is the background of our work, it is helpful to first give a brief overview of the GWR framework. The basic
formulation of GWR can be described in two equations below (Fotheringham et al., 2003).

$$y_i = \beta_0(u_i, v_i) + \sum_k \beta_k(u_i, v_i)x_{ik} + \varepsilon_i \tag{1}$$

$$\hat{\beta}_k(u_i, v_i) = (X^T W(u_i, v_i) X)^{-1} X^T W(u_i, v_i) y \tag{2}$$

In Equation 1, $y_i$ is a response variable of regression point $i$ at a location with the coordinates $(u_i, v_i)$. $x_{ik}$ is the $k^{th}$
dependent variable, and $\varepsilon_i$ denotes the error term for the $i^{th}$ observed point. A key difference between GWR and the
traditional global regression method, such as Ordinary Least Squares (OLS), is that GWR allows the coefficient $\beta_k(u_i, v_i)$
vary spatially to identify spatial heterogeneity. Equation 2 represents the GWR calibration in a matrix form. $W(u_i, v_i)$ is a
diagonal weighting matrix specific to location $i$, which is calibrated by a specified kernel function with a given bandwidth.



Every element $w_i$ in the weighting matrix reflects the impact from another observed point to the regression point. A bigger
$w_i$ value means a higher impact.

28        GWR has a strategy of spatial distance decay impact on a regression point (Brunsdon et al., 1998; Fotheringham et al.,

2003). A similar "time distance decay" strategy was also discussed in several recent GTWR models (Crespo et al., 2007;
Huang et al., 2010; Wu et al., 2014; Fotheringham et al., 2015). Yet, those models did not fully reflect the effect of time
distance decay. Sample points are observed at different time stages, and those data points closer in time distance to a
regression point have more impact on the regression point than those farther away. The time distance refers to the value
variation rate between an observed point and a regression point during a certain time interval. For example, in Fig. 1, there
are four time stages from old to new: T-s, T-q, T-p and T. Through a fitting and calibration process, the spatiotemporal
bandwidth will be fitted, and the spatiotemporal effects (weights) from observed points to a regression points at time stage T
will be calculated by a specific spatiotemporal kernel function. Then, in prediction, the value of a regression point at time
stage T can be estimated. Thus, the observed points at time stage T only have spatial effect on the regression point (Fig. 1).
There is temporal effect from data points at time stages T-p and T-q (shown as stars, pentagons and triangles in the planes of
T-p and T-q in Fig. 1), within a certain spatial bandwidth $b_{ST}$ at each time stage, to the regression point. The time distance
decay should reflect that different variations of the observed points have different temporal effects. However, as mentioned
in the previous section, many existing GTWR models have applied a strategy of time interval decay instead of time distance
decay. Consequently, they regard that all the observed points have the same temporal effect to their neighbor regression
point.

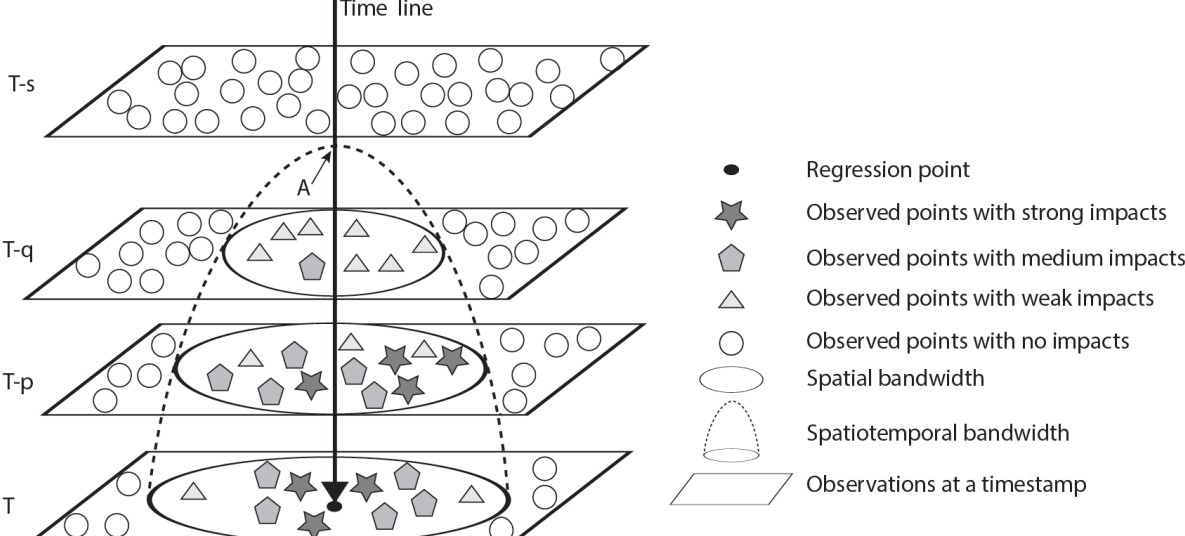


**Fig. 1**. Spatiotemporal impacts of observed points with different rates of value change on a regression point at time stage T.

Temporal bandwidth is the length of time from the intersection point A of the spatiotemporal bandwidth and the time line to

the regression point. Spatial bandwidth and spatiotemporal bandwidth are illustrated in the figure legend.


Compared to existing GTWR models, the time distance decay strategy of STWR considers the effect of different

variations of observed points through time. For example, some data points may have higher impact on the regression point,

though their spatial distance is farther than other points. Fig. 1 illustrates that the locations of some star-shape points are

farther away from the regression point than some pentagon-shape points at time stage T-p, which denotes that there exist

mixed impacts (spatial impact and temporal impact) on the regression point. The temporal impacts depend on the rate of

value variation, which is the value difference between the observed point and the regression point divided by a time interval

(e.g., [T-p, T] and [T-q, T-p] each is a time interval). If the observed time stage is too long ago or the rate of value variation

is too small, and exceeds the limit of optimized temporal bandwidth for the regression point (as shown by observations at

time stage T-s), the data points at this time stage may have no impact on the regression point. Even though some of those

data points may have huge difference in value and are close to the regression point in space, they are not within the range of





the optimized temporal bandwidth. Spatial bandwidths also vary along the time line, and usually the bandwidth gets larger
when the observation time is closer to the time stage of the regression point (Fig. 1).
**2.2 The spatiotemporal kernel function of STWR**
We assume that a set of observed points $O_{\Delta t} = \{O_{N_t}, O_{N_{t-1}}, \ldots O_{N_{t-q}} | \Delta t = [t-q, t]\}$ are collected during a certain time
interval $\Delta t$ in a study area, where $t$ represents the current time stage and $N_{t-i}, i \in \{0,1,2,\ldots,q\}(N_t = N_{t-0})$ denotes the
number of observed points at each recorded time. As the idea described above, we can borrow neighbor points in space and
their value variation during certain recent time intervals, so we can still use Equation 1 to generate local estimates. The
weight matrix $W$ in GWR usually depends on the spatial kernel (Fotheringham et al., 2015). In STWR, we need to consider
the temporal effect, so the form of $W$ is different from that in GWR. Correspondingly, we should have a spatiotemporal
kernel, which can be understood as a temporal extension based on the spatial kernel. However, if we use a multiplication
form to combine the temporal kernel and the spatial kernel (Huang et al., 2010; Wu et al., 2014; Fotheringham et al., 2015),
we will face the problem of time and space interaction as mentioned above in the Introduction section. To address that issue,
we design a weighted average form for the spatiotemporal kernel.

$$w_{ijST}^t = (1-\alpha)k_s(d_{sij}, b_{ST}) + \alpha k_T(d_{tij}, b_T), 0 \leq \alpha \leq 1 \qquad \textbf{(3)}$$

In Equation 3, $w_{ijST}^t$ is the weight at time $t$ and at the observed location $j$. $k_s$ and $k_T$ are the spatial and temporal kernel,
respectively, and they both have a value range of 0 to 1. $\alpha$ is an adjustable parameter to scale the temporal and spatial
effects, which can be optimized with the bandwidth selections. $d_{sij}$ and $d_{tij}$ are the spatial (Euclidean) and temporal
distance between the regression point $i$ and an observed data point $j$, respectively. $b_{ST}$ is the spatial bandwidth $b_S$ at a
certain time stage $T$, and $b_T$ denotes the temporal bandwidth.

79       The time distance, as mentioned above, is not the time interval but the rate of value variation between an observed point

and a regression point through a time interval. Following the time distance decay strategy in STWR, we can further derive
the temporal kernel $k_T$ as shown below.

$$w_{ij\Delta t}^t = \begin{cases} \left[ \dfrac{2}{1+exp(-\frac{|(y_{i(t)}-y_{j(t-q)})/y_{j(t-q)}|}{\Delta t/b_T})} - 1 \right], & if\ 0 < \Delta t < b_T \\ 0, & otherwise \end{cases} \qquad \textbf{(4)}$$





In Equation 4, $y_{i(t)} - y_{j(t-q)}$ is the subtraction of the regression point $i$'s observed value at $t$ from the point $j$'s observed value at $t - q$, which denotes the value change during the time interval $\Delta t$. The internal part of the exponential function is negative, in order to make the weight $w_{ij\Delta t}^t$ range from 0 to 1. The faster the value change rate is, the bigger the weight is, which means that the time impact is larger. When the time interval $\Delta t$ is out of the range $(0, b_T)$, the weight will be set to zero, which denotes that there is no impact because the observed variation is too far to affect the current moment. For example, if the price of a nearby house has changed a long time ago, it may have little or no impact on the present house price. But if the house price had a sharp change recently, it will have a big impact on the present house price. Therefore, the faster the rate of observed value changes and the shorter the time interval is, the greater the impact on the regression point will be. Compared with GTWR models, the advantage of STWR is that the temporal kernel function $k_T$ can better leverage the variation data.

To calibrate the weight value $w_{ijST}^t$, we need a spatial kernel function. The most widely used kernel functions are bi-square and Gaussian (Fotheringham et al., 2003), which are given in Equations 5 and 6, respectively.

$$Bi - square: \quad w_{ijS} = \begin{cases} \left[1 - (\frac{d_{sij}}{b_S})^2\right]^2, if \ d_{sij} < b_s \\ 0, otherwise \end{cases} \tag{5}$$

$$Gaussian: \quad w_{ijS} = exp\left[-\frac{1}{2}\left(\frac{d_{sij}}{b_S}\right)^2\right] \tag{6}$$

In Equations 5 and 6, $b_S$ is the spatial bandwidth. Derived from $b_S$ and $b_{ST}$, $b_{St}$ is the initial spatial bandwidth at the given time stage $t$ of the regression point (i.e., $t$ is the initial time for searching observed points in the past). Many functions can be specified for the change of spatial bandwidth during the time intervals. Because in most cases it will have smooth change during a certain short time interval, we assume that the spatial bandwidth changes linearly along with time, as defined bellow.

$$b_{ST} = b_{St} - tan\theta * \Delta t, \quad -\frac{\pi}{2} < \theta < \frac{\pi}{2} \tag{7}$$

In Equation 7, $tan\theta$ denotes the slope of spatial bandwidth change in correspondence to $\Delta t$, and $b_{St}$ denotes the initial spatial bandwidth at $t$. Importing Equations 4 to 7, the calibration of Equation 3 can be further derived into Equations 8 and



9, which are our spatiotemporal kernel functions in STWR. Equations 8 and 9 are based on the bi-square and Gaussian

kernel, respectively. With the STWR spatiotemporal kernel, we only need to optimize the parameters $\alpha$ and $\theta$ instead of the

spatial bandwidth $b_{ST}$. However, we shall traverse all the observed points at the initial time stage $t$ to find the optimized

spatial bandwidth $b_{St}$. Moreover, we shall also traverse all the time stages to find the optimized temporal bandwidth $b_T$.

$$w_{ijST}^{t} = \begin{cases} \left[ (1-\alpha)*\left[1-(\dfrac{d_{sij}}{b_{St}-\tan\theta*\Delta t})^2\right]^2 + \alpha*(2/(1+\exp(-\dfrac{\left|(y_{i(t)}-y_{j(t-q)})/y_{j(t-q)}\right|}{\Delta t/b_T}))-1) \right], \\ \quad if\ \Delta t < b_T,\ \ and\ \ d_{sij} < (b_{St}-\tan\theta*\Delta t) \\ 0,\ \ otherwise \end{cases}$$

(8)

$$w_{ijST}^{t} = \begin{cases} \left[ (1-\alpha)*\exp\left[-\dfrac{1}{2}\left(\dfrac{d_{sij}}{b_{St}-\tan\theta*\Delta t}\right)^2\right] + \alpha*(2/(1+\exp(-\dfrac{\left|(y_{i(t)}-y_{j(t-q)})/y_{j(t-q)}\right|}{\Delta t/b_T}))-1) \right], \\ \quad if\ \Delta t < b_T,\ \ and\ \ d_{sij} < (b_{St}-\tan\theta*\Delta t) \\ 0,\ \ otherwise \end{cases}$$

(9)

## 3 STWR in Operation

### 3.1 Bandwidth selection and parameter estimation

Some goodness-of-fit diagnostics (Loader, 1999) are widely used in general GWR-based models, such as the cross-

validation (CV) score (Cleveland, 1979; Bowman, 1984) and the Akaike Information Criterion (AIC) (Akaike, 1973;

Akaike, 1998). For STWR, we use cross-validation (CV) as the default searching criteria and we also calculate the value of a

corrected version of AIC (Hurvich et al., 1998), the AICc, which is defined bellow.

$$AIC_c = 2n\,ln(\hat{\sigma}) + n\,ln(2\pi) + n\left\{\frac{n+tr(S)}{n-2-tr(S)}\right\}$$

(10)

In Equation 10, n is the sample size, $\hat{\sigma}$ is the estimated standard deviation of the error term, and $tr(S)$ denotes the trace of

the hat matrix $S$ (Hoaglin and Welsch, 1978).



Although there is no need to optimize spatial bandwidth $b_{ST}$ of the past time stages in STWR, other parameters such as
$\alpha$ and $\theta$ need to be optimized. Also, we should give the $b_T$ and initial $b_{St}$ through trials. For more potential combinations
of these parameters for different spatiotemporal processes, a more reasonable limit and optimization procedure is hence
needed.

**3.2 Calibration of STWR**

Calibration of the STWR models can be conducted by using weighted least squares. The estimator for the coefficients at
location $(u_i, v_i)$ is shown below.

$$\hat{\beta}_t(u_i, v_i) = [(X_{O_{\Delta t}}^T W_{\Delta t}(u_i, v_i) X_{O_{\Delta t}})^{-1} X_{O_{\Delta t}} W_{\Delta t}(u_i, v_i)] y_{O_{\Delta t}} \tag{11}$$

In Equation 11, $X_{O_{\Delta t}}$ and $y_{O_{\Delta t}}$ are observed independent and dependent variables of $O_{\Delta t}$ respectively. $X_{O_{\Delta t}}^T$ is the
transpose of $X_{O_{\Delta t}}$. $W_{\Delta t}(u_i, v_i)$ denotes the spatiotemporal weight matrix for observed points at different locations to the
regression point $(u_i, v_i)$ at different time stages during $\Delta t$. For a better illustration, we show the weight matrix $W_{\Delta t}$ during
the time interval $\Delta t$ in Fig. 2. The matrix $W_{\Delta t}$ here is a bit different form the $W(u_i, v_i)$ in Equation 2. The records in the
$i^{th}$ row of $W_{\Delta t}$ are the diagonal elements in $W(u_i, v_i)$, and only no zero values are used to calibrate the coefficients $\hat{\beta}_k$ for
each regression point. Thus, each row $r$ of this hat matrix is shown below.

$$r_{it} = X_{it}(X_{\Delta t}^T W_{i\Delta t} X_{\Delta t})^{-1} X_{\Delta t} W_{i\Delta t} \tag{12}$$

In Equation 12, $X_{it}$ is the $i^{th}$ row of the matrix of independent variables at $t$. $X_{\Delta t}$ is the matrix of independent variables
during a time interval $\Delta t$, and $X_{\Delta t}^T$ is its transpose. Although the $X_{\Delta t}$ in Equation 12 is equal to the $X_{O_{\Delta t}}$ in Equation 11 in
the fitting and calibration of STWR, we distinguish $X_{O_{\Delta t}}$ from $X_{\Delta t}$ here. Because $X_{O_{\Delta t}}$ is a specific matrix of independent
variables of an observed point set $O_{\Delta t}$ during $\Delta t$, while $X_{\Delta t}$ is a general matrix of independent variables of points during
$\Delta t$. $X_{O_{\Delta t}}$ is only used for fitting and calibration of STWR, while $X_{\Delta t}$ can also be used for prediction in STWR. In other
words, we can understand $X_{O_{\Delta t}}$ as a subclass of $X_{\Delta t}$. $W_{i\Delta t}$ is the $i^{th}$ row of the weighted matrix $W_{\Delta t}$.

**3.3 Reasonable searching range and procedure of optimization**

In order to obtain the optimized $\alpha$ and $\theta$ for STWR (Equations 8 and 9), the search range should be limited. Here we use
the distance from each regression point $p_i^{(t)}$ to its $M^{th}$ nearest neighbor as the initial spatial bandwidth $b_{St}$ at $t$. The range
of $b_{St}$ is within a finite set of discrete values, because the maximum number of nearest neighbor is limited to $N_{t-i}, i \in$





$\{1,2,\ldots,q\}$ for the regression point $p_i^{(t)}$ ($N_{t-i}$ is the total number of observed points at $t-i$). We denote that value set for
$b_{St}$ as $BS_{Nt} = \{D_{k+1}, D_{k+2}, \ldots D_{N_t}\}$, in which the element $D_U, U \in \{k+1, k+2, \ldots, N_t\}$ denotes the distance from $p_i^{(t)}$ to
the $U^{th}$ nearest neighbor, and $k$ equals to the number of independent variables. Moreover, the searching range of the
temporal bandwidth $b_T$ is also limited to a finite discrete set $BT_\lambda = \{\Delta t_1, \Delta t_2, \ldots \Delta t_\lambda\}$, in which the element $\Delta t_\lambda$ is the time
interval from $t$ to $t - \lambda$.
The optimization procedure is to traverse the set $BT_\lambda$, and for each step we further traverse the set $BS_{Nt}$ to get the
optimized $\alpha$ and $\theta$ through trials. Some trials of $\theta$ may lead to no solution to Equation 11, because there might be less than
$(k+1)^{th}$ neighbors within the radius of $b_{St} - \theta\Delta t_\lambda$ from the regression point. Therefore, if it occurs at time stage $t - \lambda$,
the spatial bandwidth $b_{St} - \theta\Delta t_\lambda$ needs to be extended to the distance from its $(k+1)^{th}$ nearest neighbor to the regression
point, to guarantee the matrix in Equation 11 to be nonsingular.

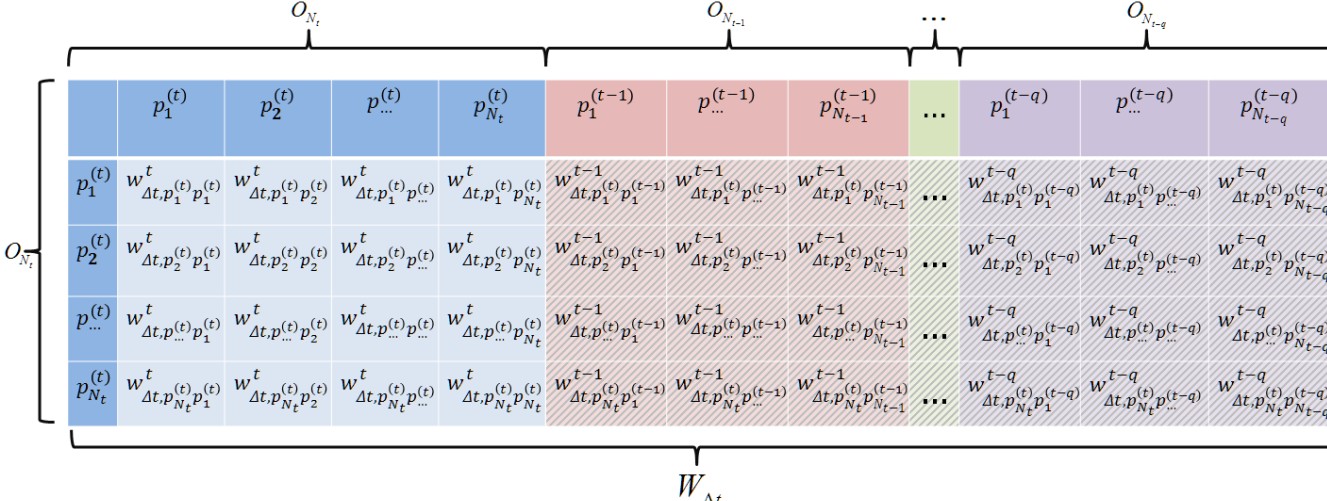


**Fig. 2.** Weight matrix $W_{\Delta t}$. The symbol $p_k^{(t-i)}, i \in \{0, 1, \ldots q\}, k \in \{1, 2, \ldots N_{t-i}\}$ denotes the $k^{th}$ observed point at $t-i$.
The symbol $w_{\Delta t, p_m^{(t)} p_n^{(t-i)}}^{t-i}, i \in \{0, 1, \ldots q\}, m \in \{1, 2, \ldots N_t\}, n \in \{1, 2, \ldots, N_{t-i}\}$ denotes the weight of the $n^{th}$ point $p_n^{(t-i)}$ at
$t-i$ to the $m^{th}$ point $p_m^{(t)}$ at $t$. The symbol $O_{N_{t-i}}, i \in \{0, 1, \ldots q\}$ denotes a set of points observed at $t-i$. $\Delta t$ denotes all
the time intervals of the weight matrix. In the central and right parts of the figure, the records with background shading
indicate weight values affected by temporal effects.






### 3.4 Steps of using STWR for prediction

In this paper, STWR is used to predicate the current values of regression points with known coordinates. The prediction
formulas of STWR are more complicated than GWR because the spatial distance is calculated directly from the regression
point to each observed data point, while the time distance between the regression point and the data points observed in the
past cannot be calculated directly. Therefore, we specify a few steps for the prediction in STWR. First, we need to have the
optimized initial spatial bandwidth $b_{St}$, the optimized $\alpha$ and $\theta$, the optimized number of time stages model used and the
fitted weight matrix. Second, all data points within the limited distance of spatial bandwidth at the latest time stage should be
found for the regression point. Third, all the temporal weights of these data points need to be retrieved from the established
weight matrix (Fig. 2). Fourth, we use these retrieved weights to calculate (e.g., use mean value or inverse distance
weighting value) the temporal weight on the regression point. Fifth, by combining with the calculated spatial weight and the
optimized $\alpha$ and $\theta$, we can calculate the spatiotemporal weight on the regression point. Then the value of the regression
point can be calculated.

### 4 Experiments with Simulated Data

### 4.1 Simulation design

To verify the performance of STWR and compare with the results of GWR and GTWR, several groups of simulated data
were used in this study to represent different types of heterogeneity in space and time. All the data and code used in the
experiments are shared on GitHub. Web links are provided at the end of this manuscript.

83        For GTWR, we only compared with the results generated by algorithms in Huang et al. (2010) and Wu et al. (2014),

because we did not find the software package of Fotheringham et al. (2015). The data generating process (DGP) and the
spatial heterogeneity are introduced here. The basic DGP is a linear model shown in Equation 1 and the study area is a
regular 25×25 lattice. We defined three initial surfaces to represent the spatial heterogeneity of parameters (Fig. 3), which
were generated by Equations 13, 14 and 15, respectively (Fotheringham et al., 2017). Through Equation 1, the two
independent variables $x_1$ and $x_2$ were initially generated randomly from the normal distribution $x_1^{initial} \sim N(100, 8)$ and





$x_2^{initial} \sim N(50,6)$, respectively. They can be set as any other values, and the mean values of both distributions may change

over time. The error term was generated from a normal distribution $\varepsilon \sim N(0,0.5)$.

$$\beta_{0(zh)}^t = 3 \qquad (13)$$

$$\beta_{1(lh)}^t = 1 + \frac{1}{12}(u,v) \qquad (14)$$

$$\beta_{2(hh)}^t = 1 + \frac{1}{324}[36 - (6 - u/2)^2][36 - (6 - \frac{v}{2})^2] \qquad (15)$$

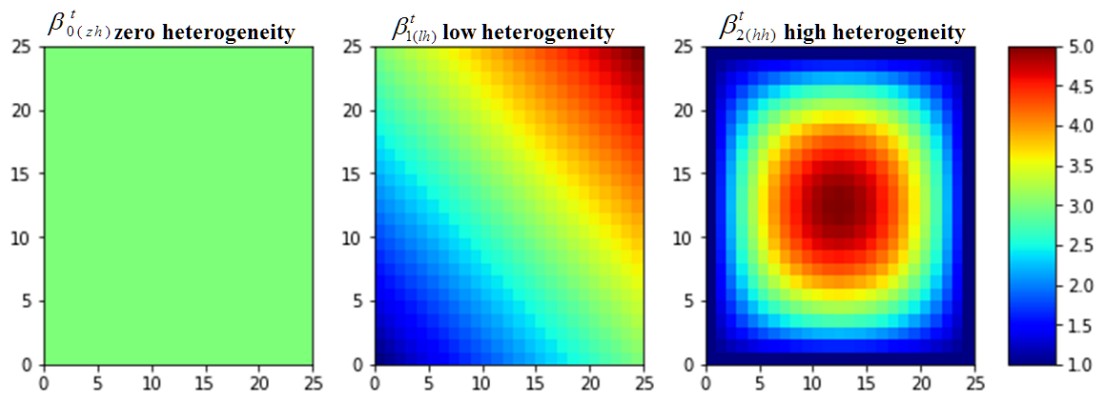

**Fig. 3.** Three simulated initial surfaces for representing spatial heterogeneity of parameters.

Several trends were designed to simulate the value change. For a better simulation, we assumed that value variation can

also be spatial heterogeneity. To distinguish from the heterogeneity of the coefficient surface, three other heterogeneity trend

functions were defined by Equations 16, 17 and 18.

$$T_1 V^{t+\Delta t} = V^t + \varphi * sin(v/4)\Delta t^{npower} \qquad (16)$$

$$T_2 V^{t+\Delta t} = V^t + \varphi * sin[1/10\pi u]\Delta t^{npower} \qquad (17)$$

$$T_3 V^{t+\Delta t} = V^t + \varphi * sin[1/6\pi(u+v)]\Delta t^{npower} \qquad (18)$$

In the above equations, $V^t$ denotes the value at time stage t, $\varphi$ is used for adjusting the magnitude of change, $\Delta t^{npower}$

denotes value change with the $n^{th}$ power of time interval, and $T_i V^{t+\Delta t}, i \in \{1,2,3\}$ denotes the $V$ value at time stage $t +$

$\Delta t$, which is the result of the $i^{th}$ trend function from the $V^t$. Fig. 4 shows these trends when $\varphi$, $V^t$, and $\Delta t^{npower}$ are set to
one.

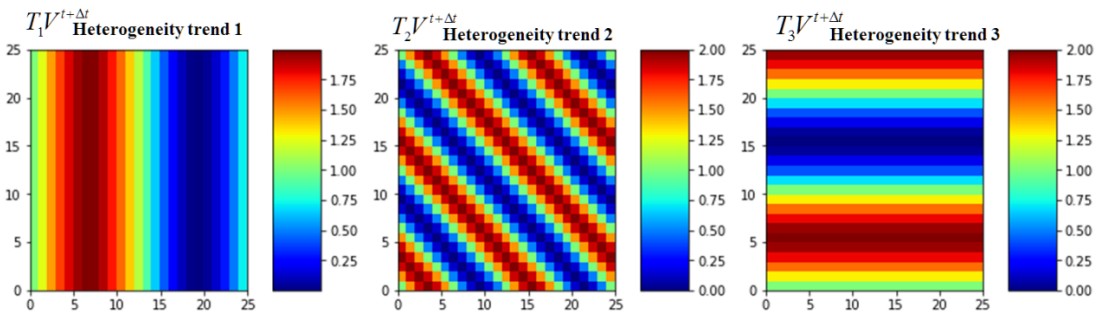

**Fig. 4.** Three heterogeneity trend surfaces.

Our goal of this experiment was to test model performance by using sample data from the simulation process at
different time. Three case studies were designed for different situations. Besides the spatial heterogeneity trends, in our
simulation design we assumed that the mean values of two independent variables $x_1$ and $x_2$ also changed over time, which
were generated by Equations 19 and 20, respectively.

$$T_1 x_m^{t+\Delta t} = x_m{}^t \pm \eta_1 * \Delta t \qquad \textbf{(19)}$$

$$T_2 x_m^{t+\Delta t} = x_m{}^t \pm \eta_2 * \Delta t \qquad \textbf{(20)}$$

In the above two equations, $x_m{}^t$ denotes the mean of an independent variable $x$ at time stage $t$, $T_i x_m^{t+\Delta t}, i \in \{1,2\}$ denotes
the mean of $x$ at time stage $t + \Delta t$, and $\eta_1$ and $\eta_2$ are two parameters for adjusting the rate of change. At each time stage
during the simulations, the independent variables $x_1$ and $x_2$ are generated by a normal distribution with new means of
$T_1 x_m^{t+\Delta t}$ and $T_2 x_m^{t+\Delta t}$, respectively.

**4.2 Results with simulated data**

We compared the results of OLS, GWR, GTWR, and STWR. A total of 333 random sample points for five time stages
$(t_0, t_1, t_2, t_3, t_4$ from old to new) were collected from the 25×25 lattice generated in the above-mentioned DGP. To simplify
the calculation process, we set $\theta$ of Equation 7 to zero. Due to the limitation of paper length, in the comparison below the





STWR results only include those generated by the spatiotemporal kernel in Equation 8. The objective is to compare the
predicted results with the true value at the latest time stage.

**4.2.1 Case study 1**

The time interval of observations in case study 1 was one unit, such as one second or one day. The value change of $x_1$ and
$x_2$ were generated by $\eta_1 = 0.5$ and $\eta_2 = 0.1$, and were affected by $T_1V$ with $\varphi = 0.5$ and $npower = 1$. This means that
$x_1$ and $x_2$ only changed slightly over time. Table 1 presents the results of the global OLS, GWR, GTWR and STWR at the
latest time stage, i.e., stage 5. It shows that the sum of squared errors (SSE) of prediction in STWR is much lower than the
other models in at least one magnitude. In addition, the AICc scores (Equation 10) also shows that STWR outperforms
GTWR and GWR. As shown in Table 1, the R2 (R-squared) value increases from 13.8% in OLS to 94.2% in GWR, 94.9%
in GTWR, and 99.3% in STWR. The estimated standard error, Sigma, reduces to 4.292 in STWR from 23.331 in GTWR.
Also, Fig. 5 shows that both the prediction surface (Y_pred) and the prediction error surface (Pred_Error) of STWR are more
accurate than those in GWR. Due to the limitation of the software package in Huang et al. (2010) and Wu et al. (2014), we
did not generate images for GTWR in Fig. 5, but the result can be seen from the Sigma value in Table 1.

**Table 1.** Results of case study 1 at time stage $t_4$.

| Time stage $t_4$ | SSE | AICc | R2 | Sigma |
|---|---|---|---|---|
| OLS | 676366.268 | 805.455 | 0.138 | |
| GWR | 45674.420 | 705.529 | 0.942 | 33.277 |
| GTWR | 40056.823 | 616.641 | 0.949 | 23.331 |
| STWR | 5761.109 | 528.860 | 0.993 | 4.293 |


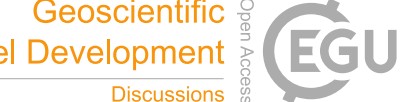

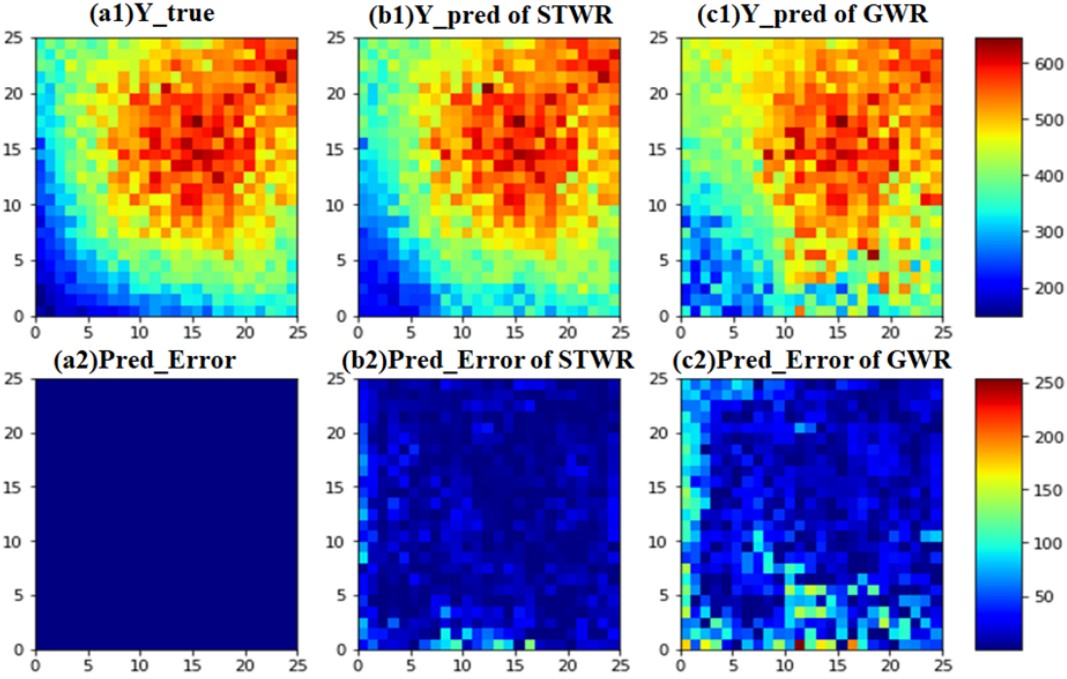

**Fig. 5.** Comparing prediction results of STWR and GWR in case study 1. Images a1, b1, and c1 are the simulation surfaces

of true Y, the predicted surface of Y by STWR, and the predicted surface of Y by GWR, respectively. Images a2, b2, and c2

are the surface of simulation error, the surface of prediction error of STWR, and the surface of prediction error of GWR,

respectively.

**4.2.2 Case study 2**

The time interval of observations in case study 2 was 10 units. The value change of $x_1$ was generated by $\eta_1 = 0.5$ and

affected by $T_3V$ with $\varphi = 0.5$ and *npower*=2. $x_2$ was generated by $\eta_2$=2 and affected by $T_2V$ with $\varphi = 1$ and *npower*

= 1, which denotes that $x_1$ and $x_2$ changed fast over time. Table 2 shows the results of the global OLS, GWR, GTWR and

STWR at the time stage 5. The SSE value in STWR is much lower than other models, and STWR has the highest R2 value

0.995. The Sigma value of STWR is 13.299, which is the lowest and less than one-fifth of the Sigma in GWR and less than

one-sixth of the Sigma in GTWR. Besides, the AICc scores show that STWR significantly outperforms GTWR and GWR.



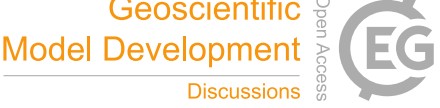

STWR utilized data from the latest three time stages to calibrate the model. The initial spatial bandwidth $b_{st}$ of STWR
was three nearest neighbors, which was smaller than the one in GWR with 15 nearest neighbors. The optimized $\alpha$ of STWR
was 0.08, which shows that the effect of used observed points to their local regression points was mainly determined by their
spatial distance. In this case, the GWR outperforms GTWR, which may due to the higher ratio of value change. Compared
with the y_true surface, the predict surface of STWR is much better than GWR (Fig. 6). For the same reason as mentioned in
case study 1, we did not generate images for GTWR in Fig. 6.

**Table 2.** Results of case study 2 at time stage $t_4$.

| Time stage $t_4$ | SSE | AICc | R2 | Sigma |
|---|---|---|---|---|
| OLS | 5085961.816 | 938.610 | 0.494 | |
| GWR | 300088.969 | 840.178 | 0.970 | 87.201 |
| GTWR | 627011.021 | 895.6621222 | 0.938 | 127.821 |
| STWR | 52688.545 | 709.573 | 0.995 | 13.299 |




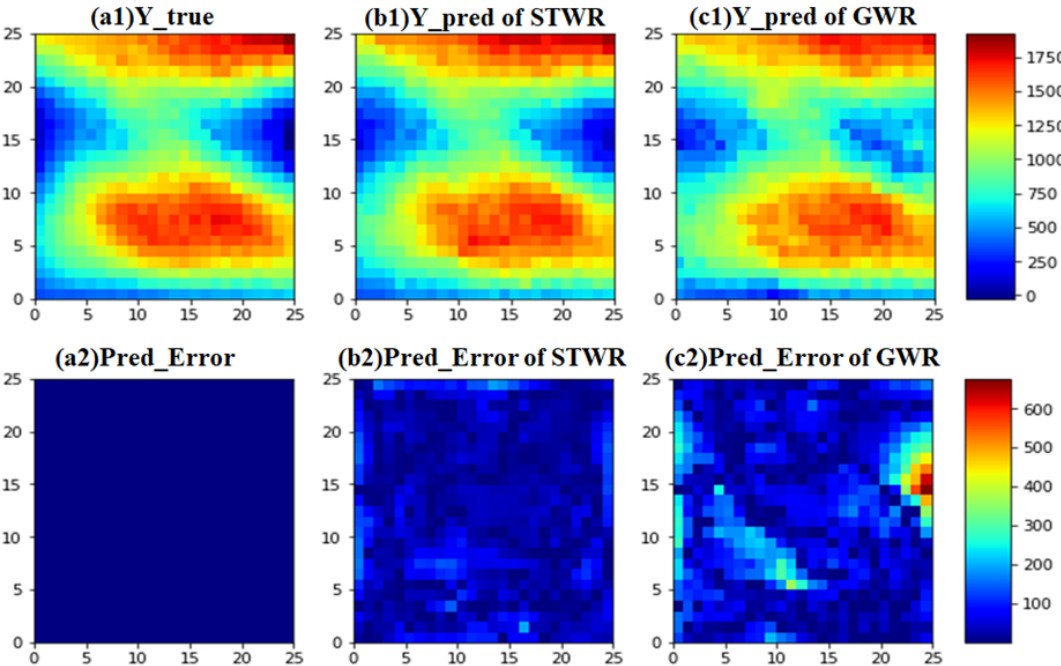

'63

**Fig. 6.** Comparing prediction results of STWR and GWR in case study 2. Images a1, b1, and c1 are the simulation surfaces

of true Y, predicted surface of Y by STWR, and predicted surface of Y by GWR, respectively. Images a2, b2, and c2 are the

surface of simulation error, the surface of prediction error of STWR, and the surface of prediction error of GWR,

respectively.

**4.2.3 Case study 3**

The time interval of observations in case study 3 was 200 units. In both case studies 1 and 2, the coefficients in Equation 1

were unchanged. In contrast, in case study 3, three surfaces of coefficients changed over time, which were generated by the

trends $T_1 V$, $T_2 V$, and $T_3 V$, respectively. The variations of coefficients were assumed to be slow. The $\varphi$ and *npower* in

each trend were set to be 0.2 and 1, respectively. Both $\eta_1$ and $\eta_2$ were set to be 0.5. The dynamic process of the three

surfaces of coefficients and the y_true surface at each time stage are shown in Fig. 7. The process in case study 3 is more

complicated than a general process, but it may be closer to reality.



**Fig. 7.** Dynamic process of three surfaces of coefficients and the y_true surface at five different time stages.

Results of these comparisons in case study 3 show that STWR outperforms both GWR and GTWR in accuracy of

model and effectiveness of simulation process (Fig. 8a). Along with the change of the coefficients and the increase of

$x_1$ and $x_2$, the R2 values of both GWR and GTWR are consistent in the five time stages, showing an overall downward

trend. But the R2 of STWR is stable and is at a high level among the five time stages. At the beginning stage $t_0$, the R2

values of the three models are similar because there are no previous observations that can be used by STWR and GTWR.





'85    The small difference among these models at $t_0$ may be caused by their different searching range of spatial bandwidth.

'86    Starting from time stage $t_1$, STWR and GTWR can borrow points from previous observations. At time stage $t_1$, STWR

'87    outperforms both GWR and GTWR, and the advantage of STWR becomes more obvious in the later stages.

'88         It may seem strange that GWR can outperform GTWR (Fig. 8), but that is reasonable for the process in case study 3.

'89    The change of this process is faster; and the time interval of observations is bigger than the previous case studies. STWR is

'90    not only able to deal with time intervals, but also to make full use of the value variation of observed points for calibration. In

'91    contrast, GTWR only uses the time interval information and all the observed points to calibrate, which may cause problems

'92    when the observed values are significantly different in spatial distribution or the time intervals are long. GTWR makes use of

'93    points from previous time stages without considering their variation, but if the actual values are quite different from previous

'94    observations at the current time stage, all the point values for the calibration of GTWR will become smooth. Thus, GWR

'95    outperforms GTWR in this situation because GWR only uses the current data points for model calibration.

'96         STWR is better for estimation than GWR and GTWR because its Sigma value is much smaller. As shown in Fig. 8b,

'97    the Sigma of STWR was half of GWR at time stage $t_1$, and even less than a third of GWR at time stage $t_4$. The results show

'98    that the advantage of STWR is obvious comparing with GWR and GTWR.

'99

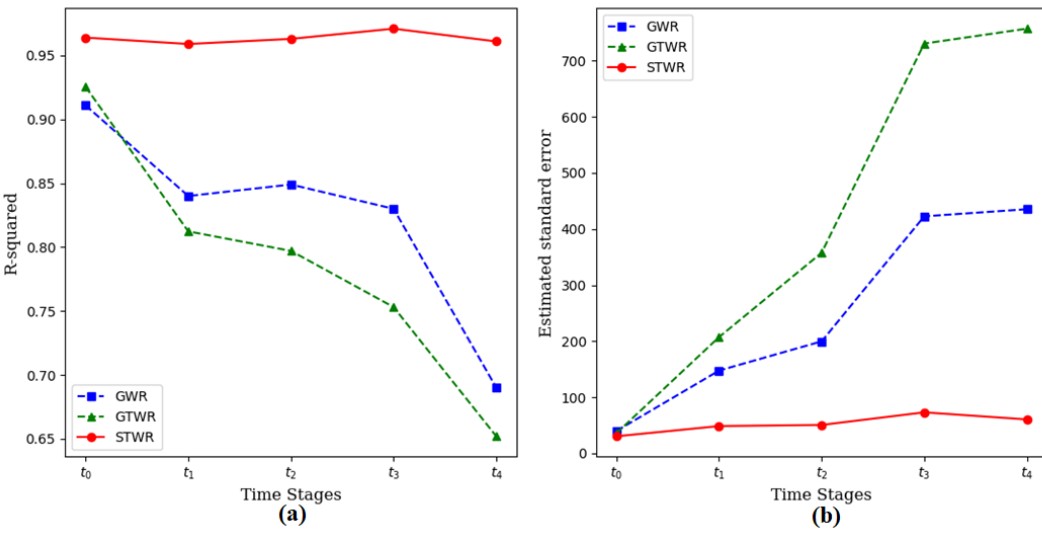

'00



·01  **Fig. 8.** Comparing and evaluating the performance of GWR, GTWR and STWR at five time stages. (a) Comparing the R2

·02  value of different models; (b) Comparing the Sigma value of different models.

·03

·04       At $t_4$, STWR used data from all the past time stages to calibrate the model, and its optimized (initial) spatial bandwidth

·05  $b_{St}$ was derived from four nearest neighbors, which was smaller than the one in GWR with 25 nearest neighbors. The

·06  optimized $\alpha$ of STWR was 0, which means that STWR only borrowed points from past time stages without considering

·07  their temporal weights to each regression point at $t_4$. The predict surfaces at time stage $t_4$ is shown in Fig. 9. The Y_pred

·08  surface of STWR is much better than GWR, especially in the middle and bottom left parts of the surface. The Pred_Error of

·09  STWR is also much lower than GWR at almost every location. In this case, the $\alpha$ of STWR at each time stage was 0, 0.96,

·10  0, 0.07, and 0, respectively. These values indicate that the temporal effects are different at each stage. They also show that

·11  the value of $\alpha$ can be adaptive to scale the temporal and spatial effects (see Equation 3).

·12       As Fig. 10 shows, the optimized bandwidths are quite different among these models, and the bandwidths of GWR and

·13  GTWR are larger than the initial bandwidth of STWR at each time stage. The optimized bandwidth for each time stage refers

·14  to an optimized number of the nearest neighbors (see Section 3.3). As GTWR considers all the nearest neighbors from

·15  different time stages, the optimized numbers of the nearest neighbors (bandwidth) grow fast, and exceed the GWR model at

·16  time stage $t_2$. However, the actual distance from the observed points to the regression points is not necessarily farther. The

·17  initial optimized numbers of the nearest neighbor of STWR are smaller than those in GWR and GTWR, which means that

·18  the initial spatial bandwidth is narrower than the bandwidth of GWR and GTWR. Nevertheless, due to the strategy of

·19  borrowing points from nearby neighbors of past observations, the total points for model calibration in STWR may still be

·20  more than GWR and GTWR. Therefore, the initial optimized numbers of the nearest neighbors in STWR are kept at a lower

·21  level, which means it is more localized than GWR in this sense.

·22



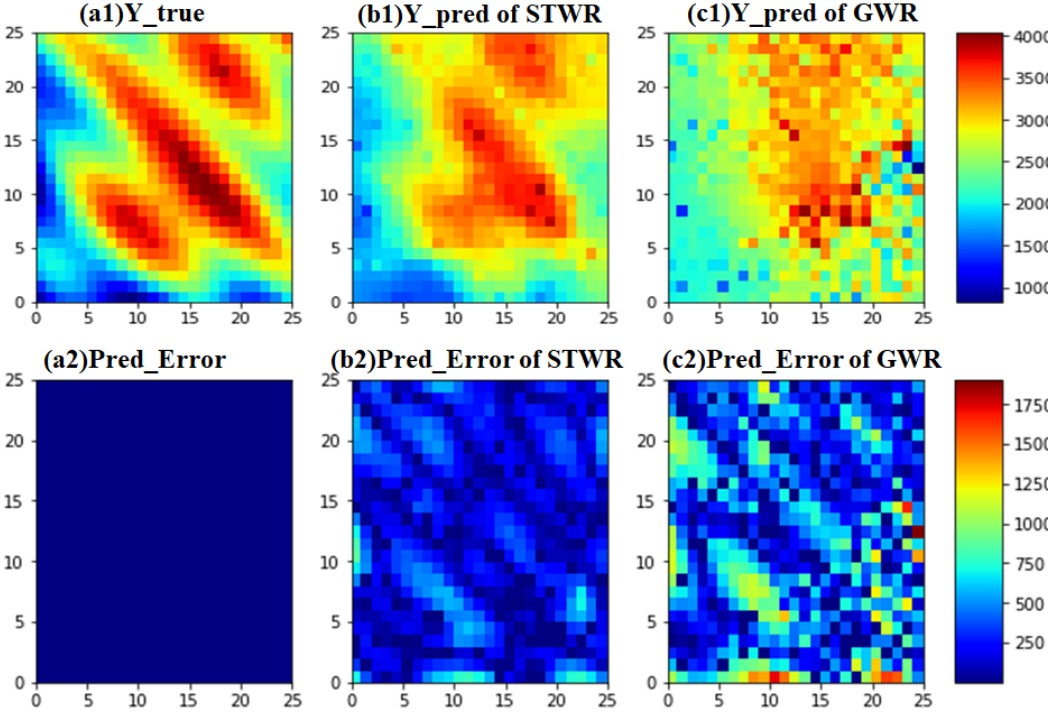

·23

**Fig. 9.** Comparing prediction results of STWR and GWR in case study 3. Images a1, b1, and c1 are the simulation surfaces

of true Y, the predicted surface of Y by STWR, and the predicted surface of Y by GWR, respectively. Images a2, b2, c2 are

the surface of simulation error, the surface of prediction error of STWR, and the surface of prediction error of GWR,

respectively.

·28

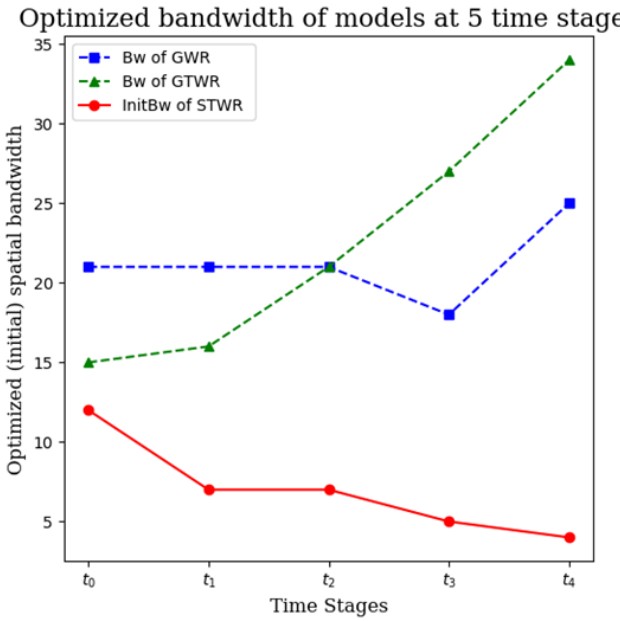

·29

**Fig. 10.** Optimized bandwidths (or initial bandwidths) of GWR, GTWR and STWR for the five time stages in case study 3.

·31

·32  **5 Experiments with Real-world Data**

·33  To further test the performance of STWR, we used data of precipitation $\delta^2$H isotopes in Northeastern United States in

·34  another case study. We chose $\delta^2$H data in three days from October 29 to 31, 2012, which have enough spatiotemporal data

·35  for the test. Here in the comparison the STWR results only include those generated by the spatiotemporal kernel in Equation

·36  8. Data and code used here are shared on Zenodo (See DOI and web links in the 'Code and data availability' section at the

·37  end of the main text of this article).

·38      In the experiments, we collected a total of 782 measurements from 116 sites located in Northeastern United States

·39  during the three-day period, and prepared the data on a daily average. The daily precipitation, mean temperature, and

·40  elevation were used as explanatory variables. The model derived from Equation 1 is represented below.

·41 $$y_i = \beta_0 + \beta_1 ppt + \beta_2 tmean + \beta_3 height + \varepsilon_i \qquad \textbf{(22)}$$

·42  In Equation 22, $ppt$ denotes the daily total precipitation (rain + melted snow), $tmean$ denotes daily mean temperature, and

·43  $height$ is the elevation value. After data preprocessing, there were 272 points for model calibration and 73 points values on



October 31, 2012. For the first day, both GTWR and STWR took no information from the past. Therefore, we only show the
results of SSE, R2 and the optimized initial neighbor (bandwidth) in the model comparisons for the second and third day (D2
and D3) in Tables 3. The SSE of STWR is the lowest at both days. GWR shows a slightly higher SSE than GTWR at D2 and
D3. The R2 of STWR is the highest at both days among these models. GWR has lower R2 than GTWR at D2, and almost the
same R2 as GTWR at D3.

49        Similar to the experiments on three simulation datasets, the result here shows that STWR outperforms GTWR and

GWR. In the experiment, the number of optimized initial neighbors of STWR was smaller than that of GWR and GTWR.
The optimized $\alpha$ of STWR was 0 at both D2 and D3. The optimized temporal bandwidths of STWR (number of time stages
model used) in both D2 and D3 were 2, which means that the STWR in this case only borrowed data points from the latest 2
time stages for D2 and D3. In the result (Table 3), an interesting part to see is that the numbers of optimized initial neighbors
of STWR are smaller than the spatial bandwidths of GWR for D2 and D3. The reason is that STWR borrowed points from
past time stages in the calculation, which led to narrower bandwidths to some extent.

57                              **Table 3.** Results of model performance with real-world data.

| Model | SSE-D2 | SSE-D3 | R2-D2 | R2-D3 | Neighbor -D2 | Neighbor -D3 |
|-------|--------|--------|-------|-------|---------|---------|
| **OLS** | 58711.528 | 52669.399 | 0.595 | 0.502 | | |
| **GWR** | 33576.400 | 33043.921 | 0.769 | 0.688 | 52 | 43 |
| **GTWR** | 32659.808 | 31967.850 | 0.775 | 0.698 | 37 | 31 |
| **STWR** | 24022.226 | 25118.096 | 0.834 | 0.763 | 16 | 16 |


59        We adopted Leave-one-out cross-validation (LOOCV) at D3 for the comparison between STWR and GWR. The

squared errors (SE) of prediction are shown in Fig. 11. The prediction results of STWR are better than GWR for most points.
The mean SE of STWR is smaller than GWR. Moreover, the SE of STWR shows a narrower regional trend, which indicates

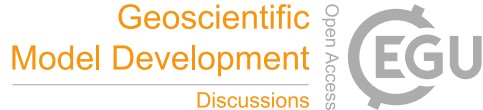

·62    that STWR is more robust than GWR. In addition, the total SSE of GWR and STWR are 50216.510 and 39724.995,

·63    respectively. Therefore, the result further validates that the quality of predication in STWR is better than GWR.

·64

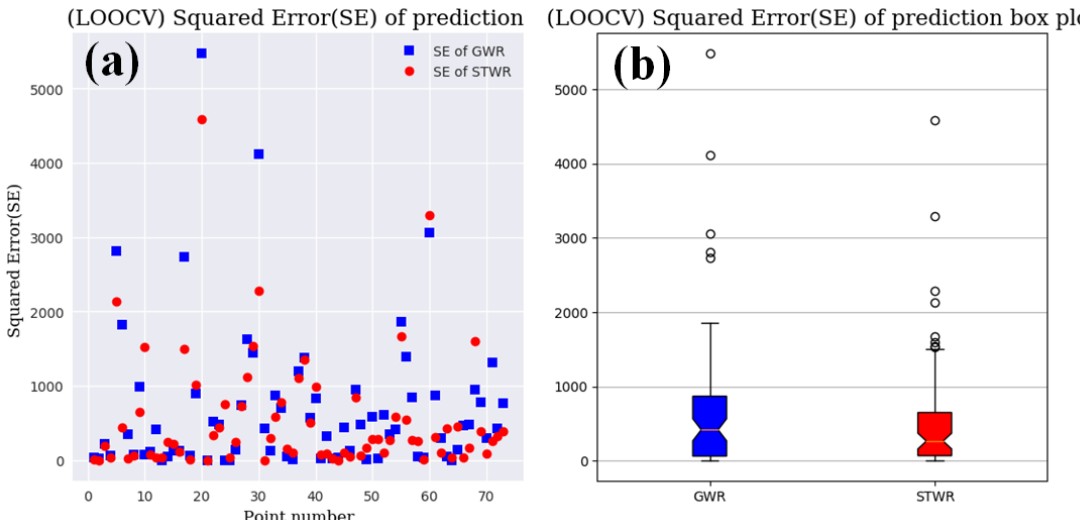

·65

·66    **Fig. 11.** LOOCV results of STWR and GWR. (a) Squared error of prediction at each point (leave out); (b) Box plot of the

·67    LOOCV results of GWR and STWR.

·68

·69         In Fig. 12, the predicted $\delta^2$H surface at D3 is broadly similar between the GWR and STWR calibrations. The

·70    percentages of explanation of variance in GWR and STWR are similar, which are 68.8% and 76.3%, respectively. However,

·71    like the experiment results with simulated data (Fig. 10), STWR has narrower initial bandwidth, which generates more

·72    localization in the predicted $\delta^2$H surface than GWR. For instance, the lower (light yellow and blue parts) or higher (orange

·73    parts) predicted values of $\delta^2$H are more concentrated in the $\delta^2$H surface of STWR than that of GWR (Fig. 12).

·74

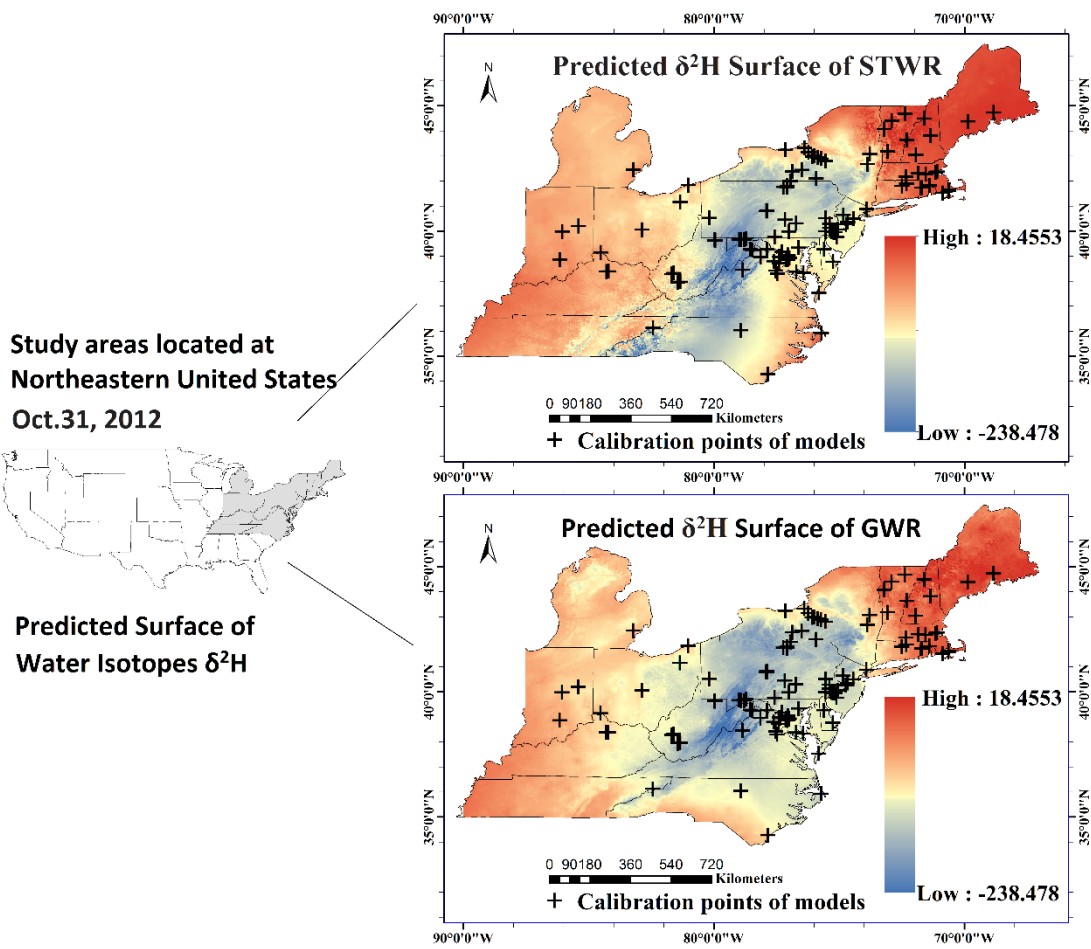

**Fig. 12.** Predicted $\delta^2 H$ surfaces of STWR and GWR at D3.

## 6 Discussion and Conclusions

Spatiotemporal data analysis is important in many scientific studies. Due to the complexity of spatiotemporal models,

spatiotemporal effect may not be fully taken into account when the temporal and spatial information is manipulated

simultaneously. In particular, the models for the effect of spatial dynamics should not be simply adapted for modeling the

effect of temporal dynamics. Although the GTWR model can borrow points from the near recent, without careful

consideration of temporal effect, the performance of GTWR may even be worse than GWR. Increasingly, many scientific

issues are not just about spatial non-stationary but involve many spatiotemporal processes. It is necessary to review the





limitation of the current spatiotemporal models and make new extensions. The aim of the STWR model developed in this
study is to advance the work and discussion in that direction.

87        The temporal kernel and the spatiotemporal kernel functions are two important contributions of STWR. The temporal

kernel in STWR applies an improved sigmoid form (see Equation 4), which is different from the methods for temporal effect
analysis in previous GTWR models. The temporal weight generated by the STWR temporal kernel is limited as a value
between 0 and 1. The spatial weight in STWR is also limited as a value between 0 and 1. The STWR spatiotemporal kernel
function has a weight adjustment parameter $\alpha$ to scale the temporal and spatial weights (Equation 3). In practice, $\alpha$ can be
obtained through optimization. This form of weighted average between temporal and spatial effects in the STWR
spatiotemporal kernel is a big improvement comparing with the multiplication form in previous GTWR models. The
advantage of the STWR spatiotemporal kernel has been proven in four case studies with both simulated and real-world
datasets.

96        Though the performance of STWR is outstanding, the models can still be further extended. A big topic is about the time

distance. In the current STWR, the time distance represents the rate of value variation between an observed point and a
regression point through a time interval. Nevertheless, we can also use time distance to represent the rate of value variation
at each observed point object through time. Note that, from an object-oriented perspective, here we differentiate the point
objects from locations, although the point objects have geospatial coordinates as part of their attributes. Following that new
definition of time distance, the $y_{i(t)} - y_{j(t-q)}$ in the STWR temporal kernel (Equation 4) can be replaced by $\Delta y_{j(t-q)}$
(value variation of an observed point object during $\Delta t$). A scenario of interest is that, the observed point objects in the past
time stages (such as those shown in Fig. 1) may move to new locations, have no value for a few time stages, or even
disappear, so the $\Delta y_{j(t-q)}$ may not exist. We can use object-based methods to address issues caused by that scenario. For
example, each point object can be assigned with a unique ID, and then the observed value of the point object at each time
stage can be retrieved by using its ID. With this new definition of time distance, the temporal weight on a regression point
object is determined by the rate of value variation of its nearby point objects. Several different scenarios for a regression
point object at current time stage $t$ are discussed here.



(1) The location of an observed point object $j$ is fixed through time (e.g., a fixed sensor). If the value of $j$ is observed at both time stages $t$ and $t-q$, then $\Delta y_{j(t-q)}$ can be calculated directly. If the value of $j$ is observed at $t$ but not observed at $t-q$, we can use interpolation to generate a value for $j$ at $t-q$. If the value of $j$ is not observed at $t$, but the variation in the past is observed, we can use prediction methods to generate a value for $j$ at $t$.

(2) The location of $j$ is not fixed through time (i.e., $j$ moves). The moving point objects can still have temporal effects to the regression point, then the $\Delta y_{j(t-q)}$ can be calculated. The spatial effect, however, depends on whether $j$ moves out of the spatial bandwidth from the regression point or not.

(3) $j$ disappears or appears at a certain time stage. If $j$ does not appear until the current time stage $t$, the $\Delta y_{j(t-q)}$ can be set to be 0. If $j$ appears in a past time stage (e.g., $t-q$) but it disappears before or at $t$, we can ignore the impact of $j$ for the regression point object.

There are other possibilities for the further improvement of STWR. The first is about the optimization of $\theta$ in the spatiotemporal kernel (Equations 8 and 9). The slope $\theta$ indicates that the variation of the spatial bandwidth is in a linear form, but it may not be a perfect solution. In many situations, the change of the spatial bandwidth over time may not be linear. The second is about making predications for future time stages. In this paper, we only predict values for points at the current time stage $t$. Extensions can be made in STWR to predict values for points in future time stages beyond $t$. The third future work is about exploring multiple spatial and temporal bandwidths of models. Different variables may have different spatial and temporal bandwidths due to their unique characteristics. Correspondingly, we may need more bandwidths to capture the different non-stationarities of those independent variables, to better represent the spatiotemporal heterogeneity.

In short, the core contribution of STWR is the clarification of the 'time distance' concept and the new temporal kernel and spatiotemporal kernel functions based on this concept. Our experiments show that STWR outperforms GWR and GTWR in analyzing and interpreting local spatiotemporal non-stationarity. We hope STWR can bring fresh ideas and new capabilities for spatiotemporal data analysis in many disciplines.

**Code and data availability**



The Python source code of STWR v1.0, the data used in the experiments and all the case studies (written in Jupyter
Notebook) were archived on Zenodo and made freely accessible via http://doi.org/10.5281/zenodo.3637689. Data source of
water isotopes δ2H is on the website: http://wateriso.utah.edu/waterisotopes/pages/spatial_db/SPATIAL_DB.html. The data
of daily precipitation and mean temperature were collected from the PRISM Climate Group
(http://www.prism.oregonstate.edu), and the elevation data were collected from the GMTED2010
(https://topotools.cr.usgs.gov/gmted_viewer/viewer.htm) at U.S. Geological Survey (USGS).

**Author Contribution.**
X.Q., X.M. and C.M. developed the algorithm, X.Q. implemented and coded the algorithm. X.Q. prepared the manuscript
with contributions from all co-authors.

**Competing interests.**
The authors declare that they have no conflict of interest.

**Acknowledgement.**
The research presented in this paper was partially supported by the National Science Foundation under Grant No. 1835717,
the China Scholarship Council under Grant No. 201807870006, and the Fujian Provincial Department of Education under
Grant No. KLA18025A. The authors thank Prof. Stewart Fotheringham and other colleagues at the Spatial
Analysis Research Center (SPARC) of Arizona State University for their insightful comments and suggestions during a
seminar about the STWR model.

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
