# Peer review of "A Spatiotemporal Weighted Regression Model (STWR v1.0) for"

_Geoscientific Model Development, 2019_

## Referee Comment (RC1) · Anonymous Referee #1 · 14 May 2020

In my opinion, the authors have done a great work. They identified a potential problem in the current spatiotemporal algorithms, and propose a new algorithm to solve this. However, I have the following concerns:

1. A possible major problem

- Not sure if I am missing something here, but the authors claim that current spatiotemporal GWR models ignore the difference in the value change of observed points during a period of time. They suggest the introduction of the rate of change in the model. They go on with the example that the faster the house price of a point changes, the stronger the temporal effect is to the house at its nearby points. To me, this makes more sense

if all observed points are measured at the same location throughout time. But in house price modeling, points are rarely measured at the same place throughout time.

- Accordingly, and following Equation (4), that distance between yi(t) and yj(t-q) will not reflect a changing rate between two houses over time because those houses are not the same. The authors should address this concern.

- The situation is different when for the four case studies used to test the algorithm because locations of observed points are the same over time. So the author may suggest the use of this new algorithm for this type of data collection.

1. Minor problems

-I think this study does not need four case studies to test the algorithm. It could be reduced to only one case study, the one with the real-world data. - The name of the journal is missing in line 79, page 31. - Reduce the number of decimals in tables - Considering that GWR provides R-square for each regression point, how should readers interpret the single the R-square shown in the tables? - Provide a possible explanation for the significant difference in the R-square values for OLS and the other methods. Something that helps readers to understand why such a big difference occurs.

---

## Author Comment (AC1) · 2 Jul 2020

Thank you for your comments. Here are our responses:

**1.For the possible major problem:**

In our current STWR algorithm, as seen in Equation (4), we use the  $y_{i(t)} - y_{i(t-q)}$  (the difference between the regression point i at time t and the observed point j at time t - q ) rather than the  $\Delta y_{j(t-q)}$  (value variation of the observed point j in  $\Delta t$  ). The main reason we use  $y_{i(t)}$  instead of  $y_{j(t)}$  to reflect the rate of change of  $y_i$  during the time interval (from t - q to t), is that the y value of the location j at t is often unavailable or may not exist at all, while the y value of the regression point i at t is known (i.e.  $y_{i(t)}$ ). Within the local spatiotemporal bandwidth, the value of  $y_{i(t)}$  is close to  $y_{j(t)}$  because both values tend to be homogeneous. As shown in the following figure, the dotted line from  $y_{j(t-q)}$  to  $y_{j(t)}$  can be approximated by the solid line from  $y_{i(t-q)}$  to  $y_{i(t)}$  within the local spatiotemporal bandwidth. When the observation point j is outside the local spatiotemporal bandwidth, there will be no such approximation. Although the value  $y_{i(t)}$  is not actual  $y_{i(t)}$ , this substitution is also valid. The reason is that both formulations can reflect the consistent temporal effect of the past observation point i on the regression point i at time t. In our STWR algorithm, we need to measure the degree of influence of the observed points at t - q (i.e.  $y_{i(t-q)}$ ) on the regression point i at t (i.e.  $y_{i(t)}$ ). The value of the difference between  $y_{i(t)}$  and  $y_{j(t-q)}$  divided by  $y_{j(t-q)}$ , which represents the numerical difference rate, can reflect the degree of temporal influence of the past observation point  $j(y_{j(t-q)})$ on the current regression point i  $(y_{i(t)})$ . Besides, we also have some ideas and suggestions about using  $\Delta y_{i(t-a)}$ in Equation (4), which is discussed in Section 6, (page 28 and 29).

**2. For the minor problems:**

We use three simulation cases and a real-world case for the reasons listing below:

(1) It can verify that this new method can be applied to different situations and is more robust than GTWR. In case 1, two independent variables  $x_1$  and  $x_2$  only changed slightly over time, and the observed time interval is short. In case 2, the  $x_1$  and  $x_2$  changed faster over time, and their observed time interval gets longer. These two cases verify that the performance of GTWR is unstable, which is sometimes better than GWR (case 1), and sometimes worse than GWR (case 2). The model performance of STWR is the best, in both case 1 and case 2, indicating that STWR is more robust than GTWR.

(2) Both case 1 and 2 assumes that three coefficient surfaces keep the same over time, but in case 3, the coefficient surfaces is assumed to vary over time. Results of the case 3 show our new algorithm STWR still outperforms GWR and GTWR models when the coefficient surfaces change over time.

(3) Through the three simulation case studies, we can draw that when the observed data changes faster over time, the outperformance of the STWR model will be more prominent than GWR and GTWR.

(4) Through the real-world case, we verified the effectiveness of our new algorithm STWR, making it more convincing.

We will add the name of the journal in line79, page 31. And we will reduce the decimal number of the AICc of GTWR in Table 2 to keep three decimal places (because some R-squares are close, keeping three digits is more convenient for comparison). Also, we will add more clear explanations and descriptions on the R-square in tables because there are many R-squares for each regression point in GWR, GTWR, and STWR. For the significant difference in the R-Square values, we will add some contents to facilitate the reader's understanding.

Thanks again for your comments.

---

## Referee Comment (RC2) · Anonymous Referee #1 · 13 Jul 2020

I am happy with the reply of the authors. I recommend this work for publication. It clearly provides new insights into temporal GWR.

Regarding the number of case studies, I understand the answer to the authors. I think that this is ultimately an editorial issue. I mean, the paper seems to be too long, but if this is Ok for the editor, it is Ok also for me.

---

## Referee Comment (RC3) · Anonymous Referee #2 · 23 Jul 2020

This paper proposes a new method named spatiotemporal weighted regression (STWR) to handle local non-stationarity in space and time. The underlying idea is interesting and meaningful and the model validation is also sufficient. Nonetheless, the method needs some more detailed explanation and discussion. My main concerns are as follows:

1. The main innovation of STWR is using the rate of value variation of the nearby observed point during the time interval to represent the time distance. However, the value variation between the estimated point and the observed points is not only influenced by the time variation but also the difference of geographical locations. How to distinguish

[Figure]

whether this effect is caused by time or space? Further, the value variation not only occurs during the time but also occurs across space. Why not also consider the value variation across space?

2. The authors indicate that the current GTWR model directly calculates the integrated spatiotemporal weights by using a multiplication of the spatial and temporal weights, which may cause underestimation of weights. This is easily misunderstood. The GTWR model also uses a scale parameter to handle the difference between time and space, which is the same as the proposed STWR model. Please correct or give more explanation.

3. As new platforms and instruments have brought increasingly massive spatiotemporal data, deep learning and neural networks have also been integrated with geostatistical models to handle spatial and temporal non-stationary relationships, such as geographically neural network regression (GNNWR), geographically and temporally neural network regression (GTNNWR). These neural network-based models can even capture the complex non-linearity in the non-stationary relationship. Some discussion or comparison between STWR with these models should be added.

---

## Author Comment (AC3) · 6 Aug 2020

The GMD manuscript system does not allow authors to submit the revised manuscript as supplement in the interactive discussion. Hope our revision report provide the necessary details about the updates made to the manuscript.

---

## Author Response (AR1)

Thanks to all the reviewers for the detailed comments and suggestions. Please check below the point-by-point response to the comments listed in the referee report, the relevant changes made to the manuscript, and a mark-up version of the manuscript with change-tracking.

===============================

**REVIEWER 1**

===============================

**Comment 1. A possible major problem**

**Not sure if I am missing something here, but the authors claim that current spatiotemporal GWR models ignore the difference in the value change of observed points during a period of time. They suggest the introduction of the rate of change in the model. They go on with the example that the faster the house price of a point changes, the stronger the temporal effect is to the house at its nearby points. To me, this makes more sense if all observed points are measured at the same location throughout time. But in house price modeling, points are rarely measured at the same place throughout time.**

**- Accordingly, and following Equation (4), that distance between yi(t) and yj(t-q) will not reflect a changing rate between two houses over time because those houses are not the same. The authors should address this concern.**

**- The situation is different when for the four case studies used to test the algorithm because locations of observed points are the same over time. So the author may suggest the use of this new algorithm for this type of data collection.**

**Reply 1.**

In our current STWR algorithm, as seen in Equation (4), we use the $y_{i(t)} - y_{j(t-q)}$ (the difference between the regression point $i$ at time $t$ and the observed point $j$ at time $t-q$ ) rather than the $\Delta y_{j(t-q)}$ (value variation of the observed point $j$ in $\Delta t$ ). The main reason we use $y_{i(t)}$ instead of $y_{j(t)}$ to reflect the rate of change of $y_j$ during the time interval (from $t-q$ to $t$), is that the $y$ value of the location $j$ at $t$ is often unavailable or may not exist at all, while the $y$ value of the regression point $i$ at $t$ is known (i.e. $y_{i(t)}$).Within the local spatiotemporal bandwidth, the value of $y_{i(t)}$ is close to $y_{j(t)}$ because both values tend to be homogeneous. As shown in the following figure, the dotted line from $y_{j(t-q)}$ to $y_{j(t)}$ can be approximated by the solid line from $y_{j(t-q)}$ to $y_{i(t)}$ within the local spatiotemporal bandwidth. When the observation point $j$ is outside the local spatiotemporal bandwidth, there will be no such approximation. Although the value $y_{i(t)}$ is not actual $y_{j(t)}$, this substitution is also valid. The reason is that both formulations can reflect the consistent temporal effect of the past observation point $j$ on the regression point $i$ at time $t$. In our STWR algorithm, we need to measure the degree of influence of the observed points at $t-q$ (i.e. $y_{j(t-q)}$) on the regression point $i$ at $t$ (i.e. $y_{i(t)}$). The value of the difference between $y_{i(t)}$ and $y_{j(t-q)}$ divided by $y_{j(t-q)}$, which represents the numerical difference rate, can reflect the degree of temporal influence of the past observation point $j$ ($y_{j(t-q)}$) on the current regression point $i$ ($y_{i(t)}$). Besides, we also have some ideas and suggestions about using $\Delta y_{j(t-q)}$ in Equation (4), which is discussed in Section 6.

**Revisions made.** To give a better explanation of the STWR model and the associated parameters, we updated the text in the second half of section 1 Introduction and the text between Equations 3 and 4. Also, we added several new paragraphs in the first half of section 6 Discussion and Conclusions to further justify the characteristics of STWR and the difference between it and other models.

[Figure]

Fig.1

**Comment 2. Minor problems**
**-I think this study does not need four case studies to test the algorithm. It could be reduced to only one case study, the one with the real-world data. - The name of the journal is missing in line 79, page 31. - Reduce the number of decimals in tables - Considering that GWR provides R-square for each regression point, how should readers interpret the single the R-square shown in the tables? - Provide a possible explanation for the significant difference in the R-square values for OLS and the other methods. Something that helps readers to understand why such a big difference occurs.**

**Reply 2.**
We use three simulation cases and a real-world case for the reasons listing below:

(1) It can verify that this new method can be applied to different situations and is more robust than GTWR. In case 1, two independent variables $x_1$ and $x_2$ only changed slightly over time, and the observed time interval is short. In case 2, the $x_1$ and $x_2$ changed faster over time, and their observed time interval gets longer. These two cases verify that the performance of GTWR is unstable, which is sometimes better than GWR (case 1), and sometimes worse than GWR (case 2). The model performance of STWR is the best, in both case 1 and case 2, indicating that STWR is more robust than GTWR.

(2) Both case 1 and 2 assumes that three coefficient surfaces keep the same over time, but in case 3, the coefficient surfaces is assumed to vary over time. Results of the case 3 show our new algorithm STWR still outperforms GWR and GTWR models when the coefficient surfaces change over time.

(3) Through the three simulation case studies, we can draw that when the observed data changes faster over time, the outperformance of the STWR model will be more prominent than GWR and GTWR.

(4) Through the real-world case, we verified the effectiveness of our new algorithm STWR, making it more convincing.

**Revisions Made.** We added the name of the journal as pointed out by the reviewer. We reduced the decimal numbers of AICc of GTWR in Table 2 to keep three decimal places (because some R-squares are close, keeping three digits is more convenient for comparison). Also, we added more clear explanations and descriptions on the R-square in tables because there are many R-squares for each regression point in GWR, GTWR, and STWR. For the significant difference in the R-Square values, we added new text to facilitate the reader's understanding.

================================

**REVIEWER 2**

================================

**Comment 1. The main innovation of STWR is using the rate of value variation of the nearby observed point during the time interval to represent the time distance. However, the value variation between the estimated point and the observed points is not only influenced by the time variation but also the difference of geographical locations. How to distinguish whether this effect is caused by time or space? Further, the value variation not only occurs during the time but also occurs across space. Why not also consider the value variation across space?**

**Reply 1.**

① We can use $y_{i(t)} - y_{j(t-q)}$ to represent the value variation between the regression point and the observation point that have time difference of $\Delta t$ ($q$). Suppose that the variation contains two parts caused by time and space, and they are $f_t(\Delta y_{j(t-q)})$ and $f_s(y_{i(t)} - y_{j(t)})$ respectively. $f_t(\Delta y_{j(t-q)})$ is not affected by spatial effects, because the location of point $j$ does not change during $\Delta t$. $f_s(y_{i(t)} - y_{j(t)})$ is not affected by temporal effects, because $y_{i(t)}$ and $y_{j(t)}$ are observed at the same time. In theory, if we get the value $y_{j(t)}$, we may determine if the variation caused by time or space, because both $f_t$ and $f_s$ need the value $y_{j(t)}$. The $y$ value of the location $j$ at $t$ (i.e. $y_{j(t)}$) is often unavailable or may not exist, we use the $y_{i(t)} - y_{j(t-q)}$ to approximate $\Delta y_{j(t-q)}$ within the local spatiotemporal bandwidth when employing the $k_T$ to calculate the temporal weights. (Please see relevant explanations in the reply 1 of the first reviewer). This may introduce some errors because of the different locations of $i$ and $j$, but the errors are limited. Consequently, the value variation between the estimated point and the observed point in different times is mainly temporal effect, the spatial effect is limited and ignored here.

② The STWR algorithm is based on the assumptions and framework of the GWR model. When calculating the spatial weights, we use the same $k_s$ employed in GWR, whose spatial impacts is calculated by the spatial distance $d_{sij}$ between $i$ and $j$. We introduce the value variation to better identify or capture the heterogeneities caused by the same time interval but different temporal effects, that is, the temporal heterogeneity of the rate of value change. The heterogeneities of this part were not considered in the previous GTWR. As for the calculation of spatial weights, the main reason that we did not consider the value variation across space is to be consistent with the GWR model, i.e. following the assumption that as long as the spatial distances between observation points to the regression point are equal, their spatial weights are the same. There may be other factors, such as anisotropy or value variation across space, that may have some additional spatial impacts on the regression point. The reasons we follow GWR's assumptions are: (a) In the optimization procedure, the model will adaptively adjust its spatial bandwidth according to the density of sampling points, and to the value variations in the space. If the value variations across the space are small, the adaptive spatial bandwidth will be large. It means that the optimization procedure already uses the information about value variation across the space. (b) If the variation $y_{i(t)} - y_{j(t)}$ was used to build a new spatial distance, which will violate the aforementioned assumption of GWR, the prediction and calibration process should be changed. Because $y_{i(t)}$ value that is required in the calculation of the new distance does not exist in prediction, spatial weights from surrounding observed points should be estimated by interpolation or other methods (just like the interpolation of temporal weights) that may bring other uncertainties or errors. Evaluating and comparing these uncertainties is not the scope of this paper in our plan. (c) If the $|y_{i(t)} - y_{j(t)}|/d_{sij}$ was used as a new spatial distance for calculating the spatial weights, we have to deal with the special case when $y_{i(t)}$ equal to $y_{j(t)}$, because the spatial kernels (such as bi-square and Gaussian) are different form the temporal kernel of STWR. In other words, if $y_{j(t-q)}$ is close or equal to $y_{i(t)}$ when employing our temporal kernel $k_T$, the output temporal weight is close or equal to 0. The underly meaning is explainable, because when the value variation gets close or equal to 0, the influence from observed point to the regression point gets weak or disappear. If $y_{i(t)}$ is close or equal to $y_{j(t)}$ when employing the bi-square or Gaussian kernel, the meaning may be difficult to understand, because when the new spatial distance $|y_{i(t)} - y_{j(t)}|/d_{sij}$ is close to 0, the output spatial weights will be large, which is inconsistent with the fact that the weaker influences it should have when the smaller value variation across space. Besides, the bi-square or Gaussian kernel have no solutions when $y_{i(t)}$ is equal to $y_{j(t)}$. If the numerator and denominator are swapped (i.e. $d_{sij}/|y_{i(t)} - y_{j(t)}|$ ), the $y_{i(t)}$ can not be equal to $y_{j(t)}$, while it is normal that $y_{j(t)}$ may be equal to $y_{i(t)}$. Therefore, if we consider combing the $y_{i(t)} - y_{j(t)}$ with $d_{sij}$ to build a new spatial distance, we may probably need to design a new appropriate spatial kernel, which requires more difficult theoretical knowledge on describing the local spatial effects.

**Revisions made.** To give a better explanation of the STWR model and the associated parameters, we updated the text in the second half of section 1 Introduction and the text between Equations 3 and 4. Also, we added several new paragraphs in the first half of section 6 Discussion and Conclusions to further justify the characteristics of STWR and the difference between it and other models.

**Comment 2. The authors indicate that the current GTWR model directly calculates the integrated spatiotemporal weights by using a multiplication of the spatial and temporal weights, which may cause underestimation of weights. This is easily misunderstood. The GTWR model also uses a scale parameter to handle the difference between time and space, which is the same as the proposed STWR model. Please correct or give more explanation.**

**Reply 2.**
The composite spatiotemporal weights might be underestimated in the current GTWR models by using the multiplication kernel. Because both outputs of the spatial kernel and the temporal kernel range from 0 to 1, and the multiplied value is never bigger than the smaller one of the spatial and temporal kernels, which means that the composite spatiotemporal impacts are never greater than the single spatial impacts and the single temporal impacts. However, the real combined spatiotemporal impacts, may be higher than the single spatial impacts or the temporal impacts, or at least may be higher than the smaller ones. Moreover, multiplication makes the weight decay faster. The role of the adjustable parameter $\alpha$ used in STWR is different from the scale parameter $\tau$ ( $\tau = \frac{u}{\lambda}$ ) in GTWR. The parameter $\alpha$ is used for adjusting the outputs of the spatial kernel $k_s$ and the temporal kernel $k_T$, which means measuring the relative strength of the spatial and temporal impacts on the regression point. However, the scale parameter $\tau$ is used for linearly adjusting the inconsistency of the distance between time and space, because of the differences of their units, scales, or metrics, etc. Specifically, GTWR uses parameters $u$ and $v$ to generate the spatiotemporal distance $d_{ij}^{ST}$ (given in the following **Equation 1**). And then substituting the $d_{ij}^{ST}$ into the spatial kernel (Gaussian), its composited weights were obtained (**Equation 2**, we use $w$ to replace the α in the original formulation, which is easier to understand in symbol). This equation, after transformation, is equal to the multiplication form of two Gaussian kernels (i.e. the spatial kernel and temporal kernel). Therefore, the scale parameter $\tau$ in GTWR only adjusts the differences between time distances and space distances, which does not change the multiplication form of the spatiotemporal kernel. In contrast, the parameter $\alpha$ in STWR (**Equation 3**) is used to adjust the effects of the two kernels $k_s$ and $k_T$, and the adjusted composite spatiotemporal weight $w_{ijST}^t$ may be larger than the smaller one of the output values of $k_s(d_{sij}, b_{ST})$ and $k_T(d_{tij}, b_T)$.

$$d_{ij}^{ST} = \lambda[(u_i - u_j)^2 + (v_i - v_j)^2] + \mu(t_i - t_j)^2 \quad (1)$$

$$
\begin{aligned}
w_{ij} &= exp\left\{-\left(\frac{\lambda\left[(u_i - u_j)^2 + (v_i - v_j)^2\right] + \mu(t_i - t_j)^2}{h_{ST}^2}\right)\right\} \\
&= exp\left\{-\left(\frac{\left[(u_i - u_j)^2 + (v_i - v_j)^2\right]}{h_S^2}\right) + \frac{(t_i - t_j)^2}{h_T^2}\right\} \\
&= exp\left\{-\left(\frac{(d_{ij}^S)^2}{h_S^2} + \frac{(d_{ij}^T)^2}{h_T^2}\right)\right\} \\
&= exp\left\{-\frac{(d_{ij}^S)^2}{h_S^2}\right\} \times exp\left\{-\frac{(d_{ij}^T)^2}{h_T^2}\right\} \\
&= w_{ij}^S \times w_{ij}^T
\end{aligned}
\quad (2)
$$

$$w_{ijST}^t = (1 - \alpha)k_s(d_{sij}, b_{ST}) + \alpha k_T(d_{tij}, b_T), 0 \le \alpha \le 1 \quad (3)$$

**Revisions Made.** We gave more explanation in the revised manuscript, please see the second last paragraph in section 1 Introduction and the text between Equations 3 and 4.

**Comment 3. As new platforms and instruments have brought increasingly massive spatiotemporal data, deep learning and neural networks have also been integrated with geostatistical models to handle spatial and temporal non-stationary relationships, such as geographically neural network regression (GNNWR), geographically and temporally neural network regression (GTNNWR). These neural network-based models can even capture the complex non-linearity in the non-stationary relationship. Some discussion or comparison between STWR with these models should be added.**

**Reply 3.**
With many successful applications of deep learning and neural network in many fields, its combinations with the traditional geospatial tools is becoming a promising research topic. Geographic neural network weighted regression (GNNWR) (Du et al., 2020) is a new attempt to combine the OLS and GWR with Artificial neural networks (ANNs). Geographic and temporal neural network regression (GTNNWR) (Wu et al., 2020) is based on the GNNWR with combing a new ANNs based method to calculate the spatiotemporal distance. Our STWR algorithm is based on the GWR with a new temporal distance and spatiotemporal kernel. There are four main differences between the GTNNWR/GNNWR and STWR: ① The basic formulation of GNNWR is defined as **Equation (4)**. The $w_0(u_i, v_i)$ and $w_k(u_i, v_i)$ denote the geographical weight of the constant coefficient $\beta_0$ and coefficient $\beta_k$, respectively. It assumed that the multiplication of $w_p(u_i, v_i)$ and $\beta_p$ is equal to $\beta_p(u_i, v_i)$ $(0 \le p \le k)$. The combined $\beta_p(u_i, v_i)$ is thought as the same as the coefficients of GWR. But in STWR and GWR, the weights and the estimated coefficients are separated. The weights mainly reflect the degree of the influences from the observed points to the regression point, while the coefficient values reflect the relationships between the independent variable and dependent variable. ② GTNNWR and GNNWR use the proposed ANNs based method (**Equation 5**) to calculate the weighted matrix, which is quite different from the kernel functions used in GWR and STWR models. Although GTNNWR and GNNWR use the idea of pointwise regression, they do not consider how to "borrow points" from nearby neighbors and do not have the concept of bandwidth. Without spatial bandwidth, all observation points in the study area may have impacts on the regression point, which might violate the Tobler's first law of geography (Tobler, 1970). It may be difficult to understand the relationships between the influence weight and the spatial distances, especially when the study area and the data amounts are large. STWR has spatial bandwidths and follows the Tobler's first law of geography, which can help analyze the affected range of local regression points. ③ The data points will be divided into training set (including validation set) and test set for the GTNNWR and GNNWR, which might require more data points. Thus, it may not be appropriate for analyzing fewer amounts of data points (data acquisitions of many geoscience processes are difficult and costly). STWR and GWR do not need to divide data points into the training set (including validation set) and test set, which requires less data points than GNNWR and GTNNWR. ④ Although GTNNWR utilizing a method named spatiotemporal proximity neural network (STPNN) to calculate the spatiotemporal distance, the obtained integrated spatiotemporal distance is lack of explanation, and it is also impossible to tell apart which parts of the calculated weight is affected by time or space. Besides, there is no concept of temporal bandwidth in GTNNWR. Thus, it cannot tell us how old the historical observation points that will have impacts on the regression point. But STWR has temporal bandwidth, and it can distinguish the strength of temporal weight and spatial weight. Therefore, we can analyze the characteristics of the local interaction of time and space according to the temporal bandwidth, spatial bandwidth, and the adjustment parameter α, etc.

$$y_i = w_0(u_i, v_i)\beta_0 + \sum_{k=1}^{p} w_k(u_i, v_i)\beta_k x_{ik} + \varepsilon_i, i = 1, 2, \dots, n \quad (4)$$

$$W_i = W(u_i, v_i) = SWNN([d_{i1}^s, d_{i2}^s, \dots, d_{in}^s]^T) \quad (5)$$

Our STWR algorithm, especially the new concept of the time distance, may also be integrated with the machine learning methods, which is our future work.

**Revisions Made.** We added the discussions on the differences between STWR and GTNNWR/GNNWR to the Section 6 Discussion and Conclusions. Please see the several new paragraphs added at the first half of section 6.

**References:**

[revised manuscript text omitted]

---

## Author Response (AR2)

Dear GMD editor,

Thank you for your comments. We revised the on the manuscript accordingly and below is a list of the revisions we made. Attached to document is a marked-up version of the latest manuscript with change tracking.

1. - page 28, line 90-95: Please elaborate a bit more on recent studies on geographic neural networks (i.e. provide more details on these studies). Otherwise, it is hard to understand why (and on what basis) you compare your method with these studies in the following paragraphs. Also note that the first sentence in this paragraph "With increasing..." is incomplete and does not provide a good introduction to a comparison with neural network models (i.e., this comparison comes out of the blue).
**Reply: We revised and extended the introduction to GNNWR and GTNNWR. Please check the updated text at lines 490-498 on page 28.**

2. - page 29, line 21: The sentence "Thus, it cannot..." is not complete. Please revise.
**Reply: This sentence was revised. Please check lines 524-525 on page 29.**

Thanks again for your time.

Sincerely,
Xiaogang (Marshall) Ma

[revised manuscript text omitted]